# KDM4 Regulates the Glycolysis of Hemocytes in the Immune Priming of *Eriocheir sinensis*

**DOI:** 10.3390/ijms252313174

**Published:** 2024-12-07

**Authors:** Xinyu Zhao, Xue Qiao, Simiao Yu, Yuhao Jin, Jixiang Niu, Jie Li, Yingmei Xu, Yuehong Yang, Lingling Wang, Linsheng Song

**Affiliations:** 1College of Life Sciences, Liaoning Normal University, Dalian 116029, China; 13122313863@163.com (X.Z.); 18853858802@163.com (Y.J.); 2Liaoning Key Laboratory of Marine Animal Immunology and Disease Control, Dalian Ocean University, Dalian 116023, China; qiaoxue@dlou.edu.cn (X.Q.); yusimiao1029@163.com (S.Y.); 17631747808@163.com (J.N.); lijie12142000@163.com (J.L.); xuyingmei1@163.com (Y.X.); yangyuehonghy@163.com (Y.Y.); 3Southern Marine Science and Engineering Guangdong Laboratory (Zhuhai), Zhuhai 519000, China; 4Liaoning Key Laboratory of Marine Animal Immunology, Dalian Ocean University, Dalian 116023, China; 5Dalian Key Laboratory of Aquatic Animal Disease Prevention and Control, Dalian Ocean University, Dalian 116023, China

**Keywords:** *Eriocheir sinensis*, immune priming, KDM4, H3K9me3, glycolysis

## Abstract

Immune priming confers a sustained, augmented response of innate immune cells to a secondary challenge, a process that is characteristically reliant on metabolic reprogramming. Recent evidence suggests that histone demethylases play essential roles in the immune priming, while its regulation role in the metabolic reprogramming remains largely unknown. In the present study, the concentration of glucose was significantly down-regulated in the hemocytes of crab *Eriocheir sinensis* after secondary stimulation with *Aeromonas hydrophila*, while the expression levels of phosphofructokinase (*Es*PFK) pyruvate kinase (*Es*PK), hexokinase-2 (*Es*HK-2) and Glucose-6-phosphate dehydrogenase (*Es*G-6-PD), along with the concentrations of lactate and the ratio of NAD^+^/NADH, were elevated. Additionally, the levels of H3K9me3 and its enrichment at the promoters of *Es*PFK and *Es*G-6-PD were significantly decreased at 7 days after *A. hydrophila* stimulation. The lysine Demethylase 4 homologue (*Es*KDM4) was observed to translocate into the nucleus of crab hemocytes after *A. hydrophila* stimulation, and its activity markedly increased after secondary stimulation with *A. hydrophila*. Following RNA interference of *Es*KDM4, there was a significant increase in H3K9me3 levels, and the enrichment of H3K9me3 at the *Es*PFK and *Es*G-6-PD promoters, as well as the concentration of glucose, in the hemocytes of crabs after secondary stimulation with *A. hydrophila*. Furthermore, mRNA transcripts of *Es*PFK and *Es*G-6-PD, as well as the concentration of lactate and ratio of NAD^+^/NADH, significantly decreased after secondary stimulation. These results suggested that *Es*KDM4 mediates the enrichment of H3K9me3 at the promoters of *Es*PFK and *Es*G-6-PD, thereby regulating glycolysis during the immune priming of crabs.

## 1. Introduction

The innate immune system exhibits features of memory, termed immune priming in invertebrates or trained immunity in vertebrates, which promotes faster and more robust responsiveness to secondary challenges [1,2]. This process is accompanied by epigenetic modification and metabolic reprogramming, with a notable increase in glycolysis, which is critical for the development of effective immune responses [3,4,5,6]. Emerging research underscores the pivotal role of histone methylation in modulating glycolysis in immune memory [7,8]. Histone lysine methyltransferase (KMT) and demethylase (KDM) play a central role in the dynamic regulation of histone lysine methylation, as well as in chromatin organization and gene expression in the immune priming [9].

Cellular metabolism reprogramming is a central process involved in the induction of immune memory [6,10]. Enhanced glycolysis and glutamine-driven tricarboxylic acid cycle have been established as important metabolic pathways for trained immunity in vertebrates [11]. For instance, in mouse models of trained immunity, there is a sustained increase in glycolytic activity and pro-inflammatory gene expression observed [12]. The trained macrophages show increased glucose consumption and high levels of glycolysis end-product lactate [13]. The glycolysis process in the trained immunity is regulated particular by rate-limiting enzymes [14]. Hexokinase-2 (HK-2) and phosphofructokinase (PFK) are both up-regulated at the transcriptional and epigenetic levels in trained immunity induced by BCG vaccination [6]. In trained hematopoietic stem and progenitor cells (HSPCs) induced by β-glucan, there is an increased rate of glycolysis, which is reflected in the increased expression of glycolytic enzymes (HK-3, PFK, and PKM) as well as the rate-limiting enzyme of the pentose phosphate pathway (G-6-PD) [15]. The increased expression of glycolytic enzymes enables the enhanced glycolytic rate, which is sustained through epigenetic changes in trained immunity [6].

Immune memory is a critical aspect of the immune system, relying on the intricate balance between epigenetic modifications and metabolism, with histone methylation being the most well-understood process [16]. Specifically, H3K4me3 modification at the promoters of pro-inflammatory cytokine (TNFA, IL-6) and aerobic glycolysis genes (HK-2, and GLUT1) was specifically induced by β-glucan training [13]. In trained human monocytes, a concurrent decline in the suppressive mark H3K9me3 was observed at the promoters of pro-inflammatory genes and those implicated in glycolytic metabolism [17,18,19]. The process of histone methylation is stringently regulated by methyltransferase enzymes (KMTs) and demethylase enzymes (KDMs) [9]. KDM4 was involved in the trained immunity induced by a western-type diet in myeloid progenitor cells of mice, which primarily functioned to demethylate on H3K9 and H3K36 [20,21]. In the β-glucan trained human monocytes, the pharmacological inhibition of the KDM4 resulted in the decreased glycolysis [22]. There is a need for further understanding of how epigenetic enzymes KDM4 regulate glycolysis in immune priming of invertebrates.

Chinese mitten crab *Eriocheir sinensis* is an important aquaculture crustacean in China, while the sustainable development of the crab industry suffered seriously as a result of the predominance of pathogenic bacteria like *Aeromonas hydrophila* [23,24]. *A. hydrophila* is a Gram-negative bacterium which is widely distributed in aquatic environments, causing a variety of diseases such as ascites disease in crabs and even resulting in death [25,26]. As crabs lack an adaptive immune response and vaccination protection, knowledge about the persistence and mechanism of immune priming is urgently needed for the development of disease management strategies. In our previous reports, the enhanced immune response and survival rate were observed in the crabs following secondary exposure to previously encountered *A. hydrophila*, which are known as “immune priming” [27,28]. However, the related mechanism in the immune priming of crabs remains largely unknown. It has been reported that KDM4 mediates the induction of trained immunity by altering H3K9me3 levels in vertebrates [22]. In the present study, the changes in glycolysis level were observed in hemocytes of crabs by secondary *A. hydrophila* stimulation, and the alteration of KDM4 activity and H3K9me3 enrichment at *Es*PFK and *Es*G-6-PD promoters after the first *A. hydrophila* stimulation, the effect of *Es*KDM4 expression on glycolysis levels were investigated to unveil the regulation mechanism of KDM4 on the glycolysis in the immune priming of *E*. *sinensis.*

## 2. Results

### 2.1. The Concentration of Glucose and Lactate, the Ratio of NAD^+^/NADH, and Transcripts of EsPFK and EsG-6-PD in the Hemocytes After Secondary Stimulation with A. hydrophila

An *A. hydrophila*-induced immune priming model was established in crabs as pervious reports [26,27]. Inactivated *A. hydrophila* (AH) was injected into crabs, with normal crab saline (Ns) injected into crabs as control. Seven days after the first injection, a secondary injection of *A. hydrophila* or normal crab saline was administered. Hemolymph was collected at 12 h after the second injection in Ns + Ns, Ns + AH, AH + Ns, AH + AH groups. After secondary stimulation with *A. hydrophila* for 12 h, the glucose concentration in the hemocytes of the AH + AH group was significantly lower compared to the Ns + AH group and the Ns + Ns group (0.61- and 0.44-fold, *p* < 0.01) (Figure 1A), while the concentration of lactate and the ratio of NAD^+^/NADH in hemocytes increased significantly in the AH + AH group (*p* < 0.05) (Figure 1B,C) at 12 h after secondary stimulation with *A. hydrophila*. The mRNA transcripts of *Es*PFK in hemocytes increased significantly in the AH + AH group, which was 3.33-fold (*p* < 0.05) of that in Ns + AH group (Figure 1D). At 12 h after secondary stimulation with *A. hydrophila*, the mRNA expression levels of *Es*HK-2 in AH + AH group was significantly increased (2.22-fold of that in Ns + AH group (*p* < 0.01)) (Figure 1E), and the mRNA expression level of *Es*PK and *Es*G-6-PD in AH + AH crabs significantly increased, which was 2.33-fold and 1.89-fold (*p* < 0.05) of that in Ns + AH group (Figure 1F,G).

### 2.2. The Level of H3K9me3 Modification and H3K9me3 Enrichment at the EsPFK and EsG-6-PD Promoters After the First Stimulation with A. hydrophila

Hemolymph was collected from the crabs at 7 d after the first injection with inactivated *A. hydrophila* (AH) or normal saline (NC as a control group). At 7 d after the first stimulation with *A. hydrophila*, the H3K9me3 level in the hemocytes of the AH group was significantly lower than that in the NC group (Figure 2A,B). The H3K9me3 modification levels of the *Es*PFK and *Es*G-6-PD promoter at 7 d after the stimulation with inactivated *A. hydrophila* were examined using chromatin immunoprecipitation followed by qPCR (ChIP-qPCR). The primers were designed to cover the promoter region of *Es*PFK and *Es*G-6-PD, within approximately 1 kilobase upstream or downstream of the transcription start site (TSS). “% of input H3K9me3” indicates the ratio of the DNA fragments of each promoter region bound by H3K9me3 to the total amount of input DNA fragments without H3K9me3 antibody pull-down. At 7 d after the first stimulation with *A. hydrophila*, the H3K9me3 enrichment at the *Es*PFK promoters decreased significantly in the AH group, which was 0.63-fold of that in the NC group (*p* < 0.01) (Figure 2C). Compared to the NC group, the H3K9me3 enrichment at *Es*G-6-PD promoter was significantly decreased in the AH group (0.44-fold, *p* < 0.01) at 7 d after the first stimulation with *A. hydrophila* (Figure 2D).

### 2.3. Expression Level of EsKDM4 mRNA in Hemocytes and Different Tissues

The full-length cDNA of *Es*KDM4 (GenBank: LOC126999276) was cloned from *E. sinensis*. As shown in Appendix A, *Es*KDM4 contains typical JmjN and JmjC domains, lacking the PHD and Tudor domain in few KDM4 family members. In the phylogenetic tree, all the KDM4s were assigned into vertebrate and invertebrate branches, in which the *Es*KDM4 first gathered with the KDM4 homologues from crustaceans, then clustered with *Ooceraea biroi*, *Bomy mori*, and *Drosophila melanogaster* KDM4s to form the invertebrate branch (Appendix A).

The *Es*KDM4 transcripts were detected in hemocytes and different tissues, including heart, brain, hematopoietic tissue (HPT), muscle, and hepatopancreas. The high mRNA expression level of *Es*KDM4 was detected in hepatopancreas and muscle, which was 27.75- and 24.12-fold of that in the heart (*p* < 0.05), respectively (Appendix A). The mRNA expression levels of *Es*KDM4 in hemocytes were 4.50-fold (*p* < 0.05) of that in the heart (Appendix A).

### 2.4. The Distribution of EsKDM4 Protein in Crab Hemocytes

The recombinant plasmid (pET-30a-*Es*KDM4) was transformed into *E. coli* Transetta (DE3) and purified by the Ni^+^ affinity chromatography. After IPTG induction for 12 h, the whole cell lysate was analyzed by 12.5% SDS-PAGE. In comparison with the cell without induction lysate (Lane 1 in Figure 3A), an evident distinct band of 57 kDa was revealed, which corresponds to the predicted molecular mass of r*Es*KDM4 with 6 × His tag (Lane 2 in Figure 3A), confirming the successful expression of r*Es*KDM4 in the DE3 host. After being purified by the Ni^+^ affinity chromatography, a single band with a molecular mass of about 57 kDa was observed (Lane 3 in Figure 3A), indicating the consistency and purity of the r*Es*KDM4. The polyclonal anti-*Es*KDM4 antibodies were prepared by an injection of r*Es*KDM4 into mice, and its specificity was examined by Western blot by using the r*Es*KDM4 protein and hemocyte protein of crabs, respectively. A unique and clear band of about 65 kDa was revealed in the hemocyte protein by Western blot, which was consistent with the predicted molecular weight of *Es*KDM4 (Figure 3B). A distinct band of about 57 kDa was observed in the Western blot assay, which was consistent with the predicted molecular weight of r*Es*KDM4 (Figure 3C). These results show the high specificity of the anti-*Es*KDM4 antibody.

The subcellular localization of *Es*KDM4 was detected by immunocytochemical assay. The *Es*KDM4 protein was observed by using its polyclonal antibody and Alexa Fluor 488-labeled goat anti-mouse lgG as the secondary antibody, which was indicated by green fluorescence signals. The nucleus of hemocytes was stained by DAPI, visible as blue fluorescence signals. In the crabs of control groups, the positive signals of *Es*KDM4 were mainly distributed in cytoplasm of hemocytes (Figure 4). After *A. hydrophila* stimulation, obvious positive signals of *Es*KDM4 were witnessed in the nucleus of crab hemocytes (Figure 4), indicating that *Es*KDM4 could be translocated into the nucleus from the cytoplasm of hemocytes after immune stimulation.

### 2.5. Transcripts and Activity of EsKDM4 After Secondary Stimulation with A. hydrophila

As a histone demethylase, KDM4 can influence the epigenetic state of immune cells by catalyzing the removal of H3K9me3 methyl groups [21]. A change in H3K9me3 levels was observed in primed crabs at 7 d after the first stimulation with *A. hydrophila*, which hinted that KDM4 may be functioning in immune priming. At 12 h after secondary stimulation with *A. hydrophila*, the mRNA expression levels of KDM4 in both the Ns + AH and AH + AH groups were significantly increased (3.08- and 2.92-fold of that in the Ns + Ns group) compared to the Ns + Ns group; however, they showed no statistically detectable difference between the two groups (Figure 5A). Interestingly, the demethylase activities of *Es*KDM4 in hemocytes after secondary stimulation were significantly up-regulated in the AH + AH group, which was 1.81- and 4.47-fold (*p* < 0.05) of that in the Ns + AH and Ns + Ns groups, respectively (Figure 5B).

### 2.6. The Level of H3K9me3 in Hemocytes and H3K9me3 Enrichment at EsPFK and EsG-6-PD Promoters in the KDM4-RNAi Crabs After Secondary Stimulation with A. hydrophila

To explore the impact of *Es*KDM4 expression on H3K9me3 level in immune priming, the level of H3K9me3 in hemocytes and H3K9me3 enrichment at *Es*PFK and *Es*G-6-PD promoters was investigated after the siRNA-*Es*KDM4 was utilized to inhibit *Es*KDM4 mRNA expression. Crabs received an injection of siRNA-KDM4, with siRNA-EGFP as a control group. At 12 h after the injection of siRNA-KDM4, the crabs received another injection of inactivated *A. hydrophila*. Subsequently, a secondary challenge with *A. hydrophila* was performed at 7 d following the initial injection. A significant decrease in *Es*KDM4 expression (approximately 80%) was observed in the siRNA-KDM4 group (Appendix A). At 12 h after the second injection of *A. hydrophila* was subjected to crabs, the expression level of *Es*KDM4 mRNA in the hemocytes of KDM4-RNAi + AA crabs was significantly repressed (0.47-fold of that in EGFP-RNAi + AA group, *p* < 0.01) (Figure 6A). Western blot revealed that the levels of H3K9me3 were significantly higher in the hemocytes of KDM4-RNAi + AA group after secondary exposure to *A. hydrophila*, which was 3.98-fold (*p* < 0.05) compared to the EGFP-RNAi + AA control group (Figure 6B,C). The enrichment of H3K9me3 at *Es*PFK and *Es*G-6-PD promoters increased significantly in the KDM4-RNAi + AA group (3.29-fold and 4.82-fold of that in the EGFP-RNAi + AA group, *p* < 0.05) (Figure 6D,F), while the mRNA expression level of *Es*PFK and *Es*G-6-PD in KDM4-RNAi + AA crabs significantly decreased, which was 0.41-fold (*p* < 0.01) and 0.38-fold (*p* < 0.05) of that in EGFP-RNAi + AA group (Figure 6E,G).

### 2.7. The Concentration of Glucose and Lactate and the Ratio of NAD^+^/NADH in the KDM4-RNAi Crabs After Secondary Stimulation with A. hydrophila

To further explore the regulation of *Es*KDM4 on glycolysis in immune priming, the concentration of glucose and lactate and the ratio of NAD^+^/NADH in the hemocytes of KDM4-RNAi crabs was measured after secondary stimulation with *A. hydrophila*. The concentration of glucose in the KDM4-RNAi + AA group was significantly increased (1.92-fold, *p* < 0.05) compared to the EGFP-RNAi + AA group (Figure 7A). The ratio of NAD^+^/NADH and the concentration of lactate decreased significantly in the KDM4-RNAi + AA group, which were 0.41- fold (*p* < 0.05) and 0.74-fold (*p* < 0.01) of that in the EGFP-RNAi + AA group, respectively (Figure 7B,C).

## 3. Discussion

Immune priming in invertebrates enables their immune systems to generate a memory response against subsequent exposures to the same or different pathogens [26]. This enhanced immune response is closely related to metabolic reprogramming, particularly the intensification of glycolysis [5,10]. In the process of metabolic reprogramming, epigenetic modifications, such as histone methylation, play a crucial role [28]. Specific histone demethylases, including KDMs, can influence the metabolic status and function of immune cells [9]. So far, there is a significant dearth of understanding of how histone demethylases regulate glycolysis in the context of invertebrate immune priming. The immune priming phenomena have been observed in the crab *E. sinensis* in our previous study, where the inactivation of *A. hydrophila* enhanced the survival rate of crabs after reinfections with the same pathogen [27]. In the present study, the stimulation of *A. hydrophila* induces an H3K9me3 enrichment at the promoters of *Es*PFK and *Es*G-6-PD, which subsequently regulates their mRNA expression and modulates glycolysis in the hemocytes of crabs following secondary stimulation with the same bacteria. KDM4 affected the H3K9me3 modification in the promoters of *Es*PFK and *Es*G-6-PD, thereby regulating glycolysis in the immune priming of crabs.

Trained immunity induces profound changes in cellular metabolic pathways particularly glycolysis in vertebrates [8]. The up-regulation of glycolysis, with increased glucose consumption and release of lactate, has been described in the innate immune memory response to different inducers [29]. In the present study, changes in glucose and lactate concentrations, as well as in the ratio of NAD^+^/NADH, were enhanced in those crab hemocytes after secondary stimulation with *A. hydrophila*. Similarly, an increase in glycolysis has been observed in monocyte–macrophage and alveolar macrophage (AM) in trained mice [15]. A metabolic switch toward aerobic glycolysis has been reported as a characteristic feature of cell activation and proliferation in monocytes [12] and activated macrophages [30]. The elevated glycolytic metabolism in the trained crabs may reprogram these cells to respond to intruding pathogens more robustly and rapidly, enhancing immune effector production and phagocytic capacity [31]. In the process of glycolysis, lactate is produced as a byproduct from the metabolism of glucose, and this process is accompanied by the interconversion of NAD^+^ and NADH, which serves a crucial role as an electron acceptor. [32]. In the trained immunity of vertebrates, the immune cell may exhibit a simultaneous rise in ratio of NAD^+^/NADH and lactate content [12]. Under steady-state conditions, immune cells have low biosynthetic activity and their energy requirements are predominantly through oxidative phosphorylation (OXPHOS) and fatty-acid oxidation (FAO). Upon activation, the energy demand of innate immune cells increases, and aerobic glycolysis, glutaminolysis, cholesterol metabolism, and fatty acid synthesis can be used to meet those additional needs [33]. Metabolic intermediates, such as acetyl-CoA, fumarate, succinate, NAD^+^, and mevalonate, which are produced as a result of this metabolic rewiring, regulate the epigenetic landscape [34]. The enhanced glycolytic rate is facilitated by the increased expression of glycolytic enzymes [6]. PFK is a major regulatory glycolytic enzyme, which catalyzes the phosphorylation of fructose-6-phosphate (F6P) to produce fructose-1,6-bisphosphate (F1,6BP), a key regulatory step in glycolysis [35]. G-6-PD catalyzes the reaction that converts glucose-6-phosphate (G6P) into 6-phosphogluconolactone in the PPP pathway [36]. In the present study, *Es*PFK and *Es*G-6-PD expression in hemocytes was significantly increased after secondary stimulation with *A. hydrophila*. A notable increase in the expression of HK2 and PFKP was also observed in peripheral blood mononuclear cells in BCG-vaccinated mice [6]. Additionally, β-glucan-trained HSPCs exhibited increased glycolysis accompanied by an elevated expression of key glycolytic enzymes, including HK-3, PFK, and PKM, as well as the regulatory enzyme of the pentose phosphate pathway, G-6-PD [15]. The increase in glycolysis observed during the process of immunological memory is also found in *Oreochromis* niloticus [37], Common carp [38], *Marsupenaeus japonicus* [39], and other aquatic animals. The shifting of cellular metabolic pathways, particularly glycolysis, resulting in heightened lactate production, is considered a characteristic of trained immunity in vertebrates [40]. Glycolysis regulation during the immune priming has stimulation-related biological limitations. The regulation of glycolysis in immune memory relies on the type of immune cells, the type and intensity of the stimulus, and the physiological state of the organism [41,42]. For example, moderate stimulation promotes the metabolic pathways that support immune cell activation and effector functions. Low to moderate doses of the stimulus typically induce metabolic shifts towards glycolysis, the pentose phosphate pathway, and glutaminolysis, providing energy and substrates for cytokine production and antimicrobial activity [35,43,44]. High doses of the stimulus may overwhelm the metabolic capacity or lead to metabolic exhaustion, impairing immune cell function and promoting immune tolerance [30].

The epigenetic mechanisms of gene expression related to immune priming in invertebrates were recently proposed, in which histone modifications have been recognized as pivotal players [37]. The modification of H3K9me3 is an evolutionarily conserved epigenetic mark, and its dynamic alterations significantly influence the rapid response capabilities of immune cells in trained immunity [6,38]. Following primary immune stimulation, gene transcription activation is correlated with changes in specific histone methylation marks, such as H3K4me3 and H3K9me3 [6]. In the present study, the H3K9me3 level was decreased at 7 d after the first stimulation with *A. hydrophila*. Our findings are consistent with recent research, in which a reduction in H3K9me3 modification was found in HSPCs in diabetes-induced trained immunity [39]. In trained human monocytes, an increase in active histone marks and a reduction in the repressive histone mark H3K9me3 were noted at the loci of pro-inflammatory genes and those associated with glycolysis process [18,19,20]. In the present study, the enrichment of H3K9me3 at the *Es*G-6-PD and *Es*PFK promoter significantly down-regulated at 7 d after the first stimulation with *A. hydrophila*. Similarly, previous studies reported the decreased H3K9me3 levels at the promoter regions of glycolytic genes, particularly PFKP and PFKFB, in the trained immunity of humans [23]. The repressive influence of H3K9me3 can diminish G-6-PD expression in certain cancer cell types, thereby modulating the glycolytic pathway and enhancing the metabolism [40]. A notable reduction in H3K9me3 levels at the promoters of glycolytic genes, including hexokinase 2, PFK, glutaminase, and glutamate dehydrogenase, was also observed in BCG-induced trained monocytes [6]. The H3K9me3 is one of the most well-known histone modifications associated with gene repression and heterochromatin, which typically leads to the repression of transcriptional activity [41]. This process may occur through the recognition and binding of H3K9me3 modifications by epigenetic enzymes, which collectively influence chromatin status and regulate the expression of *Es*PFK and *Es*G-6-PD [42]. These results indicate that the enhanced transcription of *Es*PFK and *Es*G-6-PD is associated with the down-regulation of H3K9me3 levels and the enrichment of H3K9me3 at the promoter of *Es*PFK and *Es*G-6-PD.

KDM4 serves as an essential demethylase for histone H3 at lysine 9 and 36, and it has been identified as a major regulator of immune memory responses [23,43]. KDM4 is evolutionarily conserved from budding yeast to mammals, and shares demethylase activities for H3K9me3 [44,45,46]. It has been reported that in *C. elegans*, the inhibition of KDM4 effectively restored H3K9me3 levels in the environmental exposure memory [47]. In the present study, the *Es*KDM4 activity significantly increased after secondary stimulation with *A. hydrophila*, indicating its involvement in the immune priming of crabs. Similarly, a recent study in mice revealed increased KDM4A-D expression in the myeloid progenitor cells of trained immunity [22]. KDM4 primarily functions to demethylate the methyl groups on histone H3 at lysine 9, thereby influencing the epigenetic regulation of chromatin architecture and gene expression [21]. The KDM4 plays a role in inducing trained immunity by modulating the demethylation of H3K9me3 [23]. Consistently, in this study, there was a significant increase in H3K9me3 levels after secondary stimulation with *A. hydrophila* following the RNAi of *Es*KDM4, implying the effect of *Es*KDM4 to H3K9me3 methylation during immune priming. Furthermore, after the inhibition of *Es*KDM4 expression, an enrichment of H3K9me3 at the promoters of *Es*PFK and *Es*G-6-PD was noted, along with increased transcript levels of *Es*PFK and *Es*G-6-PD in the hemocytes of crabs following secondary stimulation with *A. hydrophila*. These results indicate that *Es*KDM4 is involved in regulating H3K9me3 at the promoters of the *Es*G-6-PD and *Es*PFK genes, as well as in the transcription of these genes during immune priming. It has been reported that histone demethylase’s KDM4 family regulates glycolysis in trained immunity of mammals [23]. In the present study, glucose concentration was significantly increased, while lactate concentration and NAD^+^/NADH ratio were significantly decreased after RNAi of *Es*KDM4, suggesting the inhibition of *Es*KDM4 resulted in the weakened glycolysis process. Similarly, the inhibition of KDM4 resulted in reduced lactate production in trained monocytes, [23], and the loss of KDM5C led to decreased glucose fermentation, reduced ATP production from oxidative phosphorylation, and glycolysis [48]. The induction of glycolysis was associated with higher NAD^+^/NADH ratios, a phenomenon also observed in β-glucan-trained monocytes–macrophages [10]. The above results collectively indicated that *Es*KDM4 modulated the H3K9me3 at the *Es*PFK and *Es*G-6-PD promoters and involved in the regulation of hemocyte glycolysis in the immune priming of crabs.

In conclusion, the role of KDM4 in regulating glycolysis mechanism in the immune priming of carb hemocytes was investigated. The increased concentrations of lactate and the ratio of NAD^+^/NADH were observed in hemocytes after secondary stimulation with *A. hydrophila*, accompanied by an elevated expression of *Es*PFK, *Es*PK, *Es*HK-2, and *Es*G-6-PD, while the concentration of glucose was significantly down-regulated in the hemocytes after secondary stimulation with *A. hydrophila*. The levels of H3K9me3 and its enrichment in the promoters of *Es*PFK and *Es*G-6-PD decreased at 7 d after *A. hydrophila* stimulation. *Es*KDM4 was able to inhibit the enrichment of H3K9me3 modifications at the *Es*PFK and *Es*G-6-PD promoter, suppress glucose accumulation, promote lactate generation, and increase the NAD^+^/NADH ratio. *Es*KDM4 might affect the glycolysis of hemocytes by regulating the epigenetic modification of H3K9me3 in the immune priming of crabs.

## 4. Materials and Methods

### 4.1. Animals, Immune Stimulations, and Sample Collection

Chinese mitten crabs (*E. sinensis*) with an average weight of 20.0 g were collected from a farm in Lianyungang, Jiangsu province, China, and cultured in aerated tap water at 20 ± 2 °C for one week before the experiment. All crab experiments were performed in accordance with the approval and guidelines of the Ethics Review Committee of Dalian Ocean University.

For immune stimulation experiment, eighteen crabs were employed and divided averagely into the PBS group and the *A. hydrophila* stimulation group. The crabs received an injection with 100 μL PBS and *A. hydrophila* (1 × 10^6^ CFU mL^−1^ in 0.85% sterile saline), respectively. Nine individuals were randomly sampled from each group at 12 h post injection. Hemolymph from every three crabs was mixed as a single sample and there were three replicates for each sampling. Different tissues (heart, hepatopancreas, HPT, muscle, and brain) and hemocytes were collected from other six untreated crabs (two individuals in each parallel) to examine the *Es*KDM4 expression in different tissues (N = 3).

### 4.2. Successive Immune Stimulation of Crab E. sinensis by A. hydrophila

The bacterial stimulation experiment was conducted as previously described [27]. Thirty-six crabs were randomly assigned into four groups designated as the Ns + Ns, Ns + AH, AH + Ns, and AH + AH groups (Figure 8A). In the Ns + Ns and Ns + AH groups, the crabs received a first injection with 100 μL of normal crab saline solution, and a secondary injection with 100 μL of normal crab saline solution or a diluted suspension of *A. hydrophila* (1 × 10^6^ CFU mL^−1^) at 7 d after the first injection, respectively. In the AH + Ns and AH + AH groups, the crabs were first stimulated with 100 μL of inactivated *A. hydrophila*, and then treated with 100 μL of normal crab saline solution or 100 μL of live *A. hydrophila* as the second stimulation at 7 d after the first injection, respectively. Then, hemolymph from the four groups (Ns + Ns, Ns + AH, AH + Ns, AH + AH) was collected as three parallel samples at 12 h post the second injection, for the following detections for qPCR and Western blot (N = 3). For first stimulation experiment, the other eighteen crabs were employed and divided averagely into two groups, termed as the NC and AH groups. The crabs received an injection with 100 μL 0.85% sterile saline and *A. hydrophila* (1 × 10^6^ CFU mL^−1^ in 0.85% sterile saline), respectively. Nine individuals were randomly sampled from each group as three parallel samples at 7 d post injection (N = 3). The hemocytes were harvested according to the previous description for the following detections.

### 4.3. RNA Interference (RNAi) of EsKDM4

Specific siRNAs were designed and synthesized (GenePharma, Suzhou, China) to interfere the expression of *Es*KDM4. A total of twenty-seven crabs were randomly divided into three groups (Blank group, EGFP-RNAi group, and KDM4-RNAi + AA group) with nine individuals in each group. Each group was further divided into three biological replicates, each consisting of three individuals (N = 3). Crabs in the KDM4-RNAi + AA group received an injection of 100 μL siRNA-KDM4, and those in the EGFP-RNAi + AA group received an injection of 100 μL siRNA-EGFP, respectively. The crabs in the Blank group did not receive any treatment (Figure 8B). The crabs in the KDM4-RNAi + AA group and the EGFP-RNAi + AA group were then given a successive stimulation of *A. hydrophila* as above. The hemocytes collected from three crabs were pooled together as one sample in each group at 12 h after a secondary injection of *A. hydrophila*. The RNA interference efficiency in hemocytes was detected by examining the mRNA expression of *Es*KDM4. Hemolymph samples were collected at 12 h after secondary stimulation for the following experiment.

### 4.4. RNA Isolation, cDNA Synthesis, and Sequence Analysis

The total RNA was extracted from the crab tissues and hemocytes using Trizol reagent (Invitrogen, Carlsbad, CA, USA) according to the manufacture’s protocol. The first strand of cDNA was synthesized by using total RNA (treated with DNase I) as a template and oligo dT-adaptor as primers according to the manufacturer’s protocol (TaKaRa, Dalian, China). The cDNA mixed liquor was diluted to 1:20 and stored at −80 °C for subsequent experiments.

The protein domain was predicted with the simple modular architecture research tool (SMART) version 7.0 (http://smart.embl-heidelberg.de/ (accessed on 21 March 2024)).

### 4.5. Quantitative Real-Time PCR Analysis (qPCR)

The relative mRNA expression level of *Es*KDM4, *Es*G-6-PD, and *Es*PFK were determined by qPCR. The specific primers for *Es*KDM4 (No. XM_050861684.1), *Es*G-6-PD (No. XM_050839784.1), *Es*PFK (No. XM_050832340.1), *Es*PK (No. XM_050859497), and *Es*HK-2 (No. XM_050875291) were designed based on cDNA sequences (Table 1). *Es*β-actin (No. HM053699) was used as an internal control to verify the successful transcription and to calibrate the cDNA templates for corresponding samples. The mRNA expression was analyzed by the 2^−ΔΔCT^ method [49]. Vertical bars represent the mean ± SD (N = 3). *: *p* < 0.05, **: *p* < 0.01. ***: *p* < 0.001 (*t*-test).

### 4.6. Recombinant Expression and Purification of EsKDM4 Protein and Preparation of Its Polyclonal Antibody

PCR was performed to obtain the 1355 bp fragment of the JmjC and JmjN domain of *Es*KDM4 using primers P1 and P2 (Table 1) with *EcoR* I and *Not* I sites. The PCR products were gel-purified and cloned into pET-30a expression vector with a His-tag. The recombinant plasmid (pET-30a-*Es*KDM4) was transformed into *Escherichia coli* Transetta (DE3) (TransGen Biotech, Beijing, China). The recombinant protein of *Es*KDM4 (r*Es*KDM4) with a six-His (6 × His) tag at C-terminal was purified by Ni^+^ affinity chromatography, and desalted by extensive dialysis.

Protein induction and purification were monitored using SDS-PAGE. A bacterial suspension and purified r*Es*KDM4 in β-mercaptoethanol-containing loading buffer were heated at 100 °C for 10 min before being loaded on SDS-PAGE gels. Protein bands were visualized by staining using Coomassie Protein Stain and were subsequently imaged by Amersham Imager 600 system (GE Healthcare, CHI, IL, USA) in colorimetric mode.

Six-week-old mice were immunized with r*Es*KDM4 to acquire polyclonal antibodies. Briefly, 100 μL of r*Es*KDM4 (1 mg mL^−1^) was emulsified with 100 μL complete Freund’s adjuvant (Sigma, St. Louis, MO, USA) to immunize each mouse by subcutaneous implantation, and the remaining steps were the same as in the previous description [50]. The anti-*Es*KDM4 serum was harvested by centrifugation at 3500× *g* for 30 min to obtain the antiserum. The hemocyte protein of crabs and r*Es*KDM4 were then used to detect the specificity of anti-*Es*KDM4 antibody by using Western blot.

### 4.7. Western Blotting of H3K9me3

The efficiency and specificity of polyclonal antibodies against Histone H3 (AF0009, Beyotime, Shanghai, China) and Tri-Methyl-Histone H3 (Lys9) (H3K9me3, AF5707, beyotime, Shanghai, China) was verified by Western blot assay. The polyclonal antibody against Histone H3 and H3K9me3 (diluted at 1:500 with 5% nonfat milk in TBST) was used as the primary antibody, and the HRP-linked goat-anti-rabbit/mouse IgG (1:1000 with 5% nonfat milk, Sango Biotech, Shanghai, China) was used as the secondary antibody. The membrane was incubated in a Western lighting–ECL substrate system (Thermo Scientific, Waltham, MA, USA), and visualized by Amersham Imager 600 system (GE Healthcare, CHI, IL, USA).

### 4.8. Immunocytochemical Assay Analysis of EsKDM4 Protein in Hemocytes

An immunocytochemical assay was performed to detect *Es*KDM4 phosphorylation signals in the hemocytes at 12 h after *A. hydrophila* stimulation. The crab hemocytes were harvested and incubated in Leibovitz’s L-15 medium (Gibco, Grand Island, NY, USA) for 3 h to adhere to the surface of glass slides. The cells were incubated with 50 μL anti-*Es*KDM4 antibody (diluted 1:500 in 3% BSA) as the primary antibody at 37 °C for 1 h and incubated with Alexa Fluor 488-labeled goat anti-mouse IgG (H + L) (diluted 1:1000 in 3% BSA) as the secondary antibody for 1 h. After the final three washing cycles, the slides were mounted in buffered glycerin (50%) for observation by Laser Scan Confocal Microscope (ZEISS, Oberkochen, Germany).

### 4.9. Chromatin Immunoprecipitation (CHIP) Analysis

CHIP experiments were performed using the Chromatin Immunoprecipitation Kit (P2078, Beyotime, Shanghai, China) according to the manufacturer’s instructions. The collected hemocytes were treated with formaldehyde (1% final concentration) to crosslink the DNA and protein. Then, hemocytes were lysed in SDS lysis buffer, and the cross-linked DNA was sonicated for 10 min to obtain DNA fragments around 250 bp. The cross-linked fragmented DNA was precleared with protein agarose, and the precleared DNA was incubated with H3K9me3 antibodies (Beyotime, Shanghai, China) at 4 °C overnight. The immunoprecipitates were then incubated with protein A/G agarose, and the DNA-histone complexes with protein G agarose beads were collected. The resultant immune complexes were successively washed once with low-salt buffer, once with high-salt buffer, once with LiCl buffer, and twice with TE buffer. The cross-linked fragmented DNA was eluted using an elution buffer. The same amount of cross-linked fragmented DNA without antibody precipitation was processed in the same manner and served as an input control. The cross-linked DNA was de-crosslinked with 200 mM sodium chloride at 65 °C for 4 h, and the proteins were removed by treatment with proteinase K. The resultant DNA was extracted using the phenol/chloroform/isoamyl alcohol method. The CHIPed DNA was processed further for qPCR analysis with the primer pairs (*Es*G6PD-CHIP-F/R and *Es*PFK-CHIP-F/R) (Table 1). The data were normalized to the input and statistically analyzed with *t*-test.

### 4.10. Determination of the Glucose Concentration in Hemolymph

The glucose concentration in crab hemolymph was measured by the glucose oxidase enzymatic catalyzation method. Briefly, each sample (5 µL) was mixed with 195 µL of glucose reagent in a single well of a 96-well plate. After incubation at 37 °C for 20 min, the absorbance of the sample was measured at 550 nm with a microplate reader. A standard curve was constructed using glucose standard solution to calibrate the absorbance readings and calculate the glucose concentrations in the samples.

### 4.11. Assay of NAD^+^/NADH in Hemolymph

The intracellular NAD^+^/NADH level was measured using the NAD^+^/NADH assay kit with WST-8 (Beyotime, Shanghai, China) following the manufacturer’s instructions. The hemocytes were lysed by the addition of 200 μL pre-cooling NAD^+^/NADH lysis buffer and centrifugation at 12,000× *g* for 10 min at 4 °C. About 50–100 μL of the supernatant was moved into a tube followed by 30 min incubation at 60 °C, and then centrifugation was performed at 10,000× *g* at 4 °C for 5 min. A total of 20 μL of the supernatant was moved into 96-well plates and incubated at 37 °C for 10 min, then followed by the addition of 10 μL of color-developing solution. The ratio of NAD^+^/NADH was measured by detecting the absorbance of mixed liquid at 450 nm. The concentration of protein in the supernatant was quantified using a BCA Assay Kit (Beyotime, Shanghai, China) for normalization.

### 4.12. Assay of Lactate in Hemolymph

The lactate content in hemocytes of crabs were determined by Lactic Acid Assay Kit (Nanjing Jiancheng, Nanjing, China) according to the manufacturer’s instructions. The hemocytes were lysed by the addition of 900 μL pre-cooling PBS, followed by centrifugation at 12,000× *g* at 4 °C for 10 min. Briefly, each sample (1.25 µL) was mixed with reagent (containing 62.5 µL of LDH working solution and 12.5 µL chromogenic reagent) in a single well of a 96-well plate. After incubation at 37 °C for 10 min, 125 µL of stop buffer was added into each sample to terminate the reaction. The Lactic Acid was measured by detecting the absorbance of the mixed liquid at 530 nm. The concentration of nucleus protein in the supernatant was quantified by using a BCA Assay Kit (Beyotime, Shanghai, China) for normalization. The lactate concentration was calculated using the formula:(1)$$\mathrm{Lactate}\;\mathrm{concentration}(\mathrm{mmol}/L)=\frac{\mathrm{Sample}\;\mathrm{OD}-\mathrm{Blank}\;\mathrm{OD}}{\mathrm{Standard}\;\mathrm{OD}-\mathrm{Blank}\;\mathrm{OD}}\times C_{\mathrm{standard}}\text{÷}\mathrm{Cpr}$$

Cpr: protein concentration. C_standard:_ the standard concentration was 3 mmol/L.

### 4.13. KDM4 Enzyme Activity Assay

KDM4 demethylase activity was measured using the JMJD2/KDM4 Activity Quantification Assay Kit (ab113461, Abcam, CAM, UK) according to the manufacturer’s instructions [51]. Nuclear extract was separated from hemocytes by using a Nuclear Extraction Kit (SN0020, Solarbio, Beijing, China). Absorbance was measured at 405 nm (Tecan, Männedorf, Switzerland). The concentration of the nucleus protein in the supernatant was quantified by using a BCA Assay Kit (Beyotime, Shanghai, China) for normalization. The KDM4 activity was calculated using the formula:(2)$$\mathrm{KDM}4\;\mathrm{activity}\;(\mathrm{OD}/\min/\mathrm{mg})=\frac{(\mathrm{Sample}\;\mathrm{OD}-\mathrm{Blank}\;\mathrm{OD})}{(\mathrm{Sample}\;\mathrm{Protein}\;\mathrm{Amount}\;\times \;\min)\;}\times 1000$$

min: the incubation time (minutes) was 120 min.

## Figures and Tables

**Figure 1 ijms-25-13174-f001:**
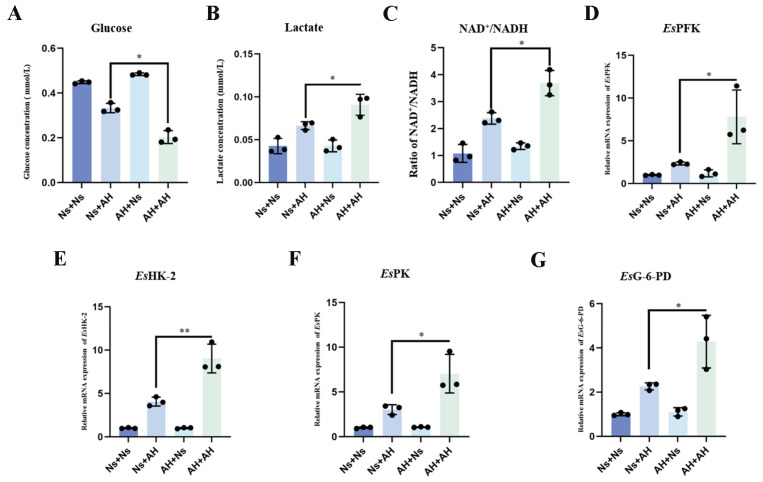
The glycolytic level in immune priming crabs. The crabs were divided in four distinct groups: Ns + Ns (control), Ns + AH (initially control followed by AH), AH + Ns (initially AH followed by control), and AH + AH (AH in both injections). Inactivated *A. hydrophila* (AH) was administered to crabs, with normal crab saline (Ns) serving as the control injection. At 7 d following the first injection, a secondary injection of *A. hydrophila* or normal crab saline was performed. Hemolymph samples were collected at 12 h post-secondary injection. (Ns, normal crab saline; AH, *A. hydrophila*). (**A**) The concentration of glucose in hemocytes was determined using a glucose oxidase assay. (**B**) Hemocyte lactate was determined by colorimetric method. (**C**) The ratio of NAD^+^/NADH in hemocytes was measured using a NAD^+^/NADH assay kit with WST-8. (**D**–**G**) Transcripts of *Es*PFK, *Es*PK, *Es*HK-2, and *Es*G-6-PD in crab hemocytes were detected by qPCR at 12 h after secondary stimulation. *Es*β-actin was used as an internal control. Vertical bars represent the mean ± SD (N = 3). *: *p* < 0.05 (*t*-test). (*Es*PFK, phosphofructokinase; *Es*G-6-PD, glucose-6-phosphate dehydrogenase; *Es*PK, pyruvate kinase; *Es*HK-2, hexokinase-2).

**Figure 2 ijms-25-13174-f002:**
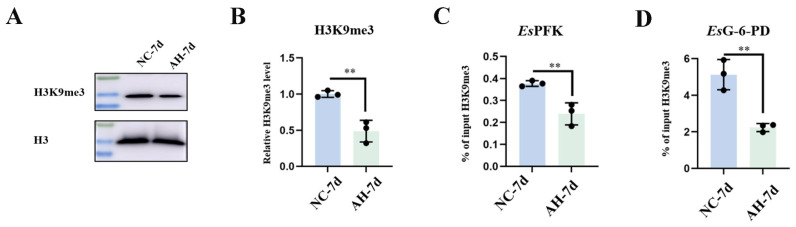
The level of H3K9me3 and its enrichment at the *Es*PFK and *Es*G-6-PD promoters at 7 d after the stimulation with *A. hydrophila.* The crabs were treated with either inactivated *A. hydrophila* or normal crab saline as a control. Seven days after the first injection, hemocytes were collected (NC-7d, normal crab saline-injected crab; AH-7d, inactivated *A. hydrophila*-injected crab). (**A**,**B**) Western blot analysis of H3K9me3 modification levels in hemocytes at 7 d after the stimulation with *A. hydrophila*. The specific antibody against H3K9me3 was used to detect H3K9me3 level. H3 was used as a loading control. (**C**) The H3K9me3 enrichment at the *Es*PFK promoters was analyzed by ChIP-qPCR. (**D**) The H3K9me3 enrichment at *Es*G-6-PD promoters was analyzed by ChIP-qPCR. The % of input H3K9me3 refers to the percentage of H3K9me3 positive-signal DNA relative to the total amount of chromatin present in the input sample. The input sample, representing 1% of the total chromatin before immunoprecipitation, served as a reference to calculate the relative enrichment. Vertical bars represent the mean ± SD (N = 3). **: *p* < 0.01 (*t*-test).

**Figure 3 ijms-25-13174-f003:**
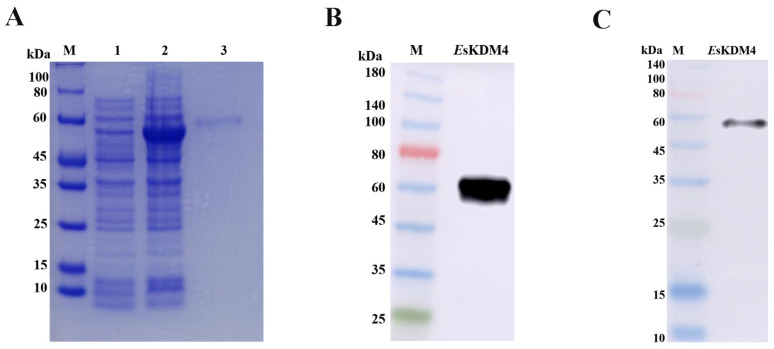
The SDS-PAGE and Western blotting of *Es*KDM4. (**A**) The SDS-PAGE analysis of the recombinant *Es*KDM4 expressed in an *E. coli* system. The recombinant plasmid (pET-30a-*Es*KDM4) was transformed into *E. coli* Transetta (DE3) and purified by the Ni^+^ affinity chromatography. After IPTG induction for 12 h, the whole cell lysate was analyzed by 12.5% SDS-PAGE. Lane M: protein molecular standard; lane 1: negative control for r*Es*KDM4 (without induction); lane 2: induced r*Es*KDM4 (the supernatant of IPTG-induced DE3 lysate); lane 3: purified r*Es*KDM4. (**B**) The specificity for the polyclonal anti-*Es*KDM4 antibody was analyzed by Western blotting by using crab hemocyte protein. (**C**) Western blotting based on the sample of lane 2, assaying the specificity of the polyclonal anti-*Es*KDM4.

**Figure 4 ijms-25-13174-f004:**
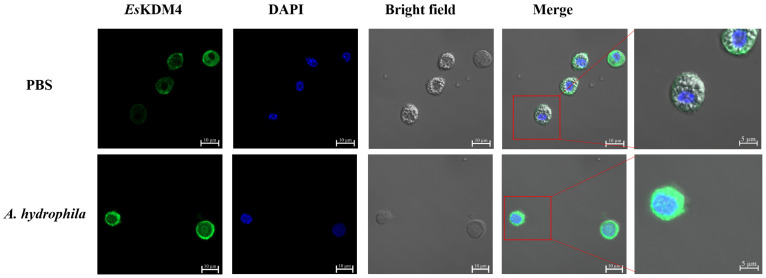
The subcellular localization of *Es*KDM4 in hemocytes. Immunocytochemical assay was performed to analyze the subcellular localization of *Es*KDM4 in hemocytes of *E. sinensis*. The subcellular localization was detected with the anti-*Es*KDM4 as the primary antibody and Alexa Fluor 488-labeled Goat Anti-Mouse IgG (H + L) as the secondary antibody (green). Nucleus staining with DAPI was shown in blue.

**Figure 5 ijms-25-13174-f005:**
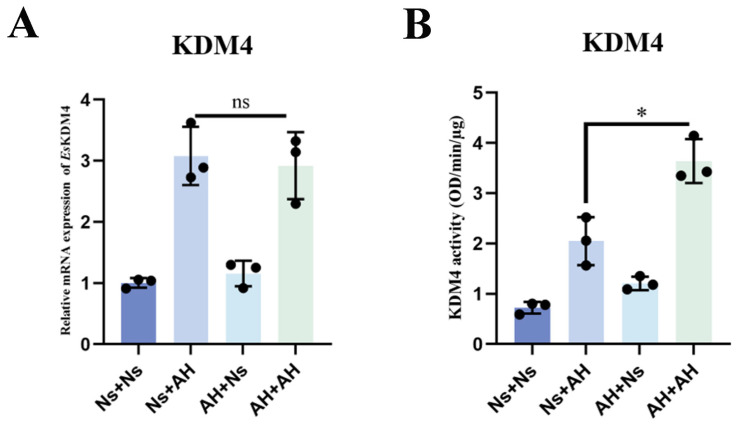
The mRNA expression levels and enzymes activity of *Es*KDM4 in immune priming crabs. The crabs individually received an injection of 100 µL of normal crab saline (Ns) or 100 µL of inactivated *A. hydrophila* (AH), with Ns serving as the control. Seven days after the initial injection, a secondary injection of either *A. hydrophila* or normal crab saline was administered. Hemolymph was collected at 12 h following the second injection from four groups: Ns + Ns, Ns + AH, AH + Ns, and AH + AH. (**A**) The relative mRNA expression level of KDM4 was detected by qPCR. *Es*β-actin was used as an internal control. (**B**) The KDM4 enzyme activity was examined using the hemocytes’ nuclear extract. Vertical bars represent the mean ± SD (N = 3). *: *p* < 0.05, ns: no significant difference (*t*-test).

**Figure 6 ijms-25-13174-f006:**
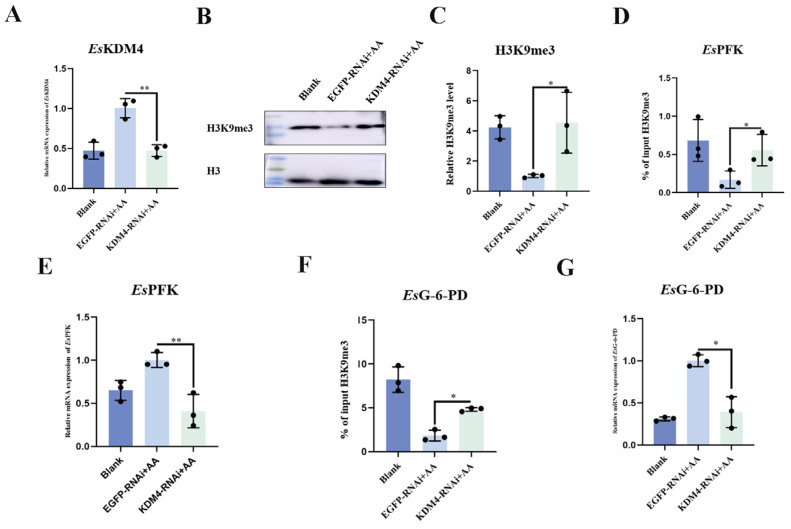
The level of H3K9me3 and its enrichment at *Es*PFK and *Es*G-6-PD promoters in KDM4-RNAi crabs after secondary stimulation with *A. hydrophila*. The twenty-seven crabs were randomly divided into three groups (Blank group, EGFP-RNAi + AA group, and KDM4-RNAi + AA group) with nine individuals in each group. Crabs in the KDM4-RNAi + AA group received an injection of 100 μL siRNA-KDM4, while those in the EGFP-RNAi + AA group received an injection of 100 μL siRNA-EGFP, respectively. The crabs in the Blank group did not receive any treatment. The crabs in the KDM4-RNAi + AA group and the EGFP-RNAi + AA group were then given inactivated *A. hydrophila* treatment. Seven days after the injection with inactivated *A. hydrophila*, a secondary injection of *A. hydrophila* was carried out. Hemolymph samples from three crabs were pooled together for each group at 12 h following the second *A. hydrophila* injection. AA, a successive injection of *A. hydrophila*. (**A**) The mRNA expression of *Es*KDM4 in hemocytes was analyzed by qPCR. *Es*β-actin was used as an internal control. (**B**) The H3K9me3 level in *Es*KDM4-RNAi + AA crab was examined following secondary stimulation with *A. hydrophila* using Western blot analysis. The specific antibody against H3K9me3 was used, with H3 as a loading control. (**C**) The statistical analysis of H3K9me3 level using Image J (version 9.4.0.). *: *p* < 0.05. (**D**,**E**) The enrichments of H3K9me3 at *Es*PFK and *Es*G-6-PD promoters in the hemocytes of KDM4-RNAi crabs + AA following secondary stimulation with *A. hydrophila*. The “% of input H3K9me3” refers to the percentage of H3K9me3 positive-signal DNA relative to the total amount of chromatin present in the input sample. Chromatin was immunoprecipitated using a specific antibody against H3K9me3, followed by qPCR to quantify the precipitated DNA. The input sample was served as a reference to calculate the relative enrichment. (**F**,**G**) The relative mRNA expression levels of *Es*PFK and *Es*G-6-PD were detected by qPCR. *Es*β-actin was used as an internal control. Vertical bars represent the mean ± SD (N = 3). *: *p* < 0.05, **: *p* < 0.01 (*t*-test).

**Figure 7 ijms-25-13174-f007:**
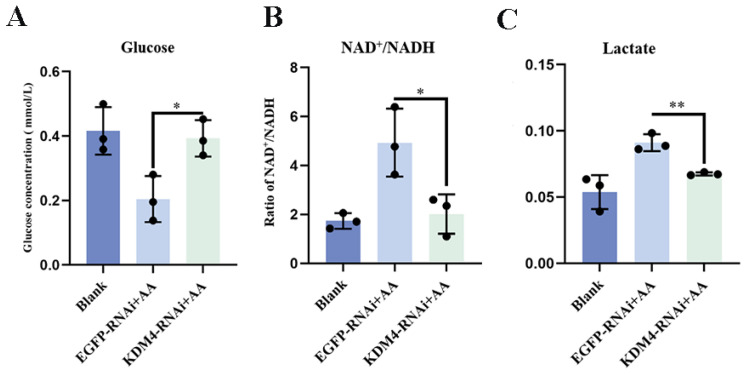
The glycolytic level in KDM4-RNAi crabs after secondary stimulation with *A. hydrophila*. The twenty-seven crabs were randomly divided into three groups (Blank group, EGFP-RNAi + AA group, and KDM4-RNAi + AA group) with nine individuals in each group. Crabs were treated with either siRNA-KDM4 or siRNA-EGFP injection, with siRNA-EGFP as control. After 12 h post injection of siRNA, inactivated *A. hydrophila* was injected into the crabs of the KDM4-RNAi + AA and EGFP-RNAi + AA groups. Subsequently, seven days after the initial injection of inactivated *A. hydrophila*, a secondary injection of *A. hydrophila* was carried out. AA, a successive injection of *A. hydrophila*. (**A**) The glucose concentration in hemocytes was determined using the glucose oxidase method. (**B**) Hemocyte lactate levels were determined by the colorimetric method. (**C**) The ratio of NAD^+^/NADH in hemocytes was measured using a NAD^+^/NADH assay kit with WST-8. Vertical bars represent the mean ± SD (N = 3). *: *p* < 0.05, **: *p* < 0.01 (*t*-test).

**Figure 8 ijms-25-13174-f008:**
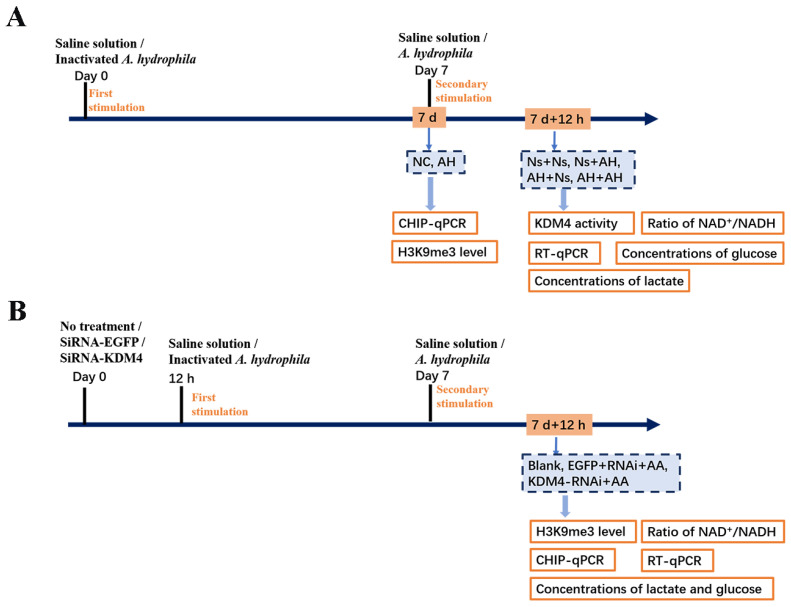
A schematic representation of the experimental design. (**A**) *A. hydrophila* -induced trained immunity in *E. sinensis.* Inactivated *A. hydrophila* (1 × 10^6^ CFU mL^−1^) was injected into crabs; the same volume of crab saline was used as control. Seven days after the first injection, a second injection of *A. hydrophila* (100 μL, 1 × 10^6^ CFU mL^−1^) or saline was administered. Hemolymph was collected at 7 d after the first injection (NC group; AH group) and 12 h after the second one (Ns + Ns, Ns + AH, AH + Ns, and AH + AH groups). (**B**) RNA interference (RNAi) of *Es*KDM4 followed by two stimulations of *A. hydrophila.* Twenty-seven crabs were randomly divided into three groups (Blank, EGFP-RNAi + AA, and KDM4-RNAi + AA); those in the Blank group did not receive any treatment, and the others received an injection of siRNA (1 OD) of EGFP, and KDM4 in PBS, respectively. Twelve hours after the injection of siRNAi in the EGFP-RNAi + AA and KDM4-RNAi + AA groups, the crabs received the first stimulation of inactivated *A. hydrophila* (1 × 10^6^ CFU mL^−1^). At 7 d after the first stimulation of *A. hydrophila*, each crab received a secondary injection with *A. hydrophila*. Hemolymph samples were collected at 12 h after secondary stimulation for the following experiment. Ns, normal crab saline; AH, *A. hydrophila*.

**Table 1 ijms-25-13174-t001:** The primers used in this study.

Primers	Sequence (5′-3′)
Clone primers	
*Es*KDM4-F	GCAATGACTTCGGTTTTCAA
*Es*KDM4-R	ATCTGGAGTTTCTTTATGAGGT
RT primers	
*Es*PFK-RT-F	AGATTACGGAGGAGGAGAGC
*Es*PFK-RT-R	TGCCTCACACGCAATACTAC
*Es*G6PD-RT-F	CACCTGGAATCGGGACAACA
*Es*G6PD-RT-R	TCCATCCCTACCACGTCCAT
*Es*PK-RT-F	ATCAGCAAGATCGAGAACCA
*Es*PK-RT-R	TCAACGGGAACCTCAATACC
*Es*HK2-RT-F	CGGCCTATTGTTTGGAGGGT
*Es*HK2-RT-R	AGACACACTCACAGGCATGG
*Es*KDM4-RT-F	GCCAAGATCATCCCACCTCC
*Es*KDM4-RT-R	AGCCTTGAACTCCTTGACCG
*Es*β-actin-RT-F	TCATCACCATCGGCAATGA
*Es*β-actin-RT-F	TTGTAAGTGGTCTCGTGGATG
Recombination protein primers	
P1 (Forward)	CGCGGATCCATGATGGGGGATCAGCCC
P2 (Reverse)	CCCAAGCTTGGAACCGTCTCCCTTCAGC
Primer for CHIP	
*Es*PFK-CHIP-F	CCGCCATGTAGTCGATGAAT
*Es*PFK-CHIP-R	TTTAGGTGGCCACACATCAC
*Es*G6PD-CHIP-F	GAGAGTTGCCAGATTGTCGT
*Es*G6PD-CHIP-R	TGTTGATCTCACCTTCCCCT
Primer for RNA interference	
siRNA-*Es*KDM4-F	GCCACUAAGCACAGCACAUTT
siRNA-*Es*KDM4-R	AUGUGCUGUGCUUAGUGGCTT

## Data Availability

Data is contained within the article and Appendix A.

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
