# Peer review of "KDM4 Regulates the Glycolysis of Hemocytes in the Immune Priming of Eriocheir sinensis"

_ijms, 2024, doi:10.3390/ijms252313174_

Round 1

Reviewer 1 Report

Comments and Suggestions for Authors

In this study a change of glycolysis level and the KDM4 activity after a second injection of  A. hydrophila was observed. Several other enzymes and molecules were also effected in their concentration during this second injection of this bacterium. In all these experiments maybe at least one experiments to clearly show that animals have been primed and survived longer may be included. The Figure legends must be more detailed since now this reviewer is unable to understand the results in the figures unless going back to M&M? More important,  it is nearly impossible to understand how the experiments have been performed since there are too many abbreviations not properly explained. In all Figures dot blots should be used so that one can see the exact variation between samples! Please also detail that true biological replicates have been made! In all silencing experiments some information about degree of silencing should be included. Otherwise this manuscript is of interest.

What is descended?

In Figure 2 what does % of input mean? Is this a quantitative measure or? This Figure shows parts of % and one wonders whether such low percent is quantifiable?

In Figure 3D this reviewer assumes that the polyclonal antibody was used against lane 2 proteins in Figure #A. If not this has to be done.

Figure 4 must be enlarged so that it is possible to see exact localization!

Give absorbance on Y-axis in Fig. 5B. If absorbance is from 2 to 4 as shown in this Figure this is not correct since at those values there is no linear relationship! The experiments have to be re-done with lover absorbance values!

The obtained values in Figure 6 are sometimes % of input and sometimes Relative mRNA expression and Relative protein expression. Very complicated for a reviewer and reader to grasp what the authors aim to show. The Figure legends and M&M do not help much to interpret these results. This reviewer though wonders whether the authors consider that these minor changes do have an affect on these animals and if so what?

Fig.7C what is mmol/gramport on the Y-axis? 

How many animals were used in this experiment in Fig.7? Why are they different measures of concentrations of glucose and lactate in these determination? This makes interpretation of data very difficult. 

The ratio of NAD7NADH is high in EGFP silenced animals and at the same time lactate is high. Is this a plausible correlation?

Comments on the Quality of English Language

English must be edited

Author Response

Dear Editor and Reviewers,

Thanks for the thorough review and insightful comments provided on our manuscript. Those comments are all valuable and very helpful for revising and improving our paper, as well as the important guiding significance to our researches. The revised parts are marked with red color in the revised manuscript. Here is a summary of the revisions and responses.

Response to reviewer

In this study a change of glycolysis level and the KDM4 activity after a second injection of A. hydrophila was observed. Several other enzymes and molecules were also effected in their concentration during this second injection of this bacterium. In all these experiments maybe at least one experiments to clearly show that animals have been primed and survived longer may be included. The Figure legends must be more detailed since now this reviewer is unable to understand the results in the figures unless going back to M&M? More important, it is nearly impossible to understand how the experiments have been performed since there are too many abbreviations not properly explained. In all Figures dot blots should be used so that one can see the exact variation between samples! Please also detail that true biological replicates have been made! In all silencing experiments some information about degree of silencing should be included. Otherwise this manuscript is of interest.

Response: We appreciate the reviewer’s comments and suggestions. The figure legends have been revised, and the methods of treatment have been included in the results section for better understanding. The results of siRNA-KDM4 interference efficiency have been added to line 204, which are shown in Supplementary Figure S3.

We have revised the figure legend to include dot blots for all figures and described biological replicates details in Materials and Methods, allows for a clearer visualization of the exact variations between samples.

  1. What is descended?

Response: Thank you for pointing out the inappropriate use of the term "descended." We have revised the sentence as follows (Line124). After secondary stimulation with A. hydrophila for 12 h, the glucose concentration in the haemocytes of the AH+AH group was significantly lower compared to the Ns+AH group and the Ns+Ns group.

  1. In Figure 2 what does % of input mean? Is this a quantitative measure or? This Figure shows parts of % and one wonders whether such low percent is quantifiable?

Response: We appreciate the reviewer’s comments and suggestions. Here, % of input refers to the percentage of H3K9me3 positive-signal DNA relative to the total amount of chromatin present in the input sample. In Figure 2 CHIP-qPCR experiments, "% of input" is a common method of data normalization used to assess the enrichment of specific DNA fragments in an immunoprecipitation (IP) experiment relative to the total amount of chromatin input before the chromatin immunoprecipitation process. The input sample, representing 1% of the total chromatin before immunoprecipitation, served as a reference to calculate the relative enrichment [1]. For better comprehension, the vertical axis is changed to "% of input H3K9me3". The corresponding modifications have been added to Figure legends.

Quantitative real-time PCR was used for the detection and precise quantification of specific DNA sequences. As reported in previous studies, the enrichment H3K9me3 at Nrf2 promoter (% of input)is approximately 0.2%, which is consistent with our experimental data [2].

  1. In Figure 3D this reviewer assumes that the polyclonal antibody was used against lane 2 proteins in Figure #A. If not this has to be done.

Response: Thanks very much for the reviewer’s suggestion. In Figure 3, the induced rEsKDM4 protein (the whole cell lysate) was shown in lane 2. Western blot analysis of polyclonal antibody against-EsKDM4 specificity was used the induced rEsKDM4 protein (as shown in lane 2 Figure 3A) and have updated the data in Figure 3C accordingly.

  1. Figure 4 must be enlarged so that it is possible to see exact localization!

Response: Thanks for the reviewer’s comments and suggestion. We have enlarged the corresponding cells and updated in Figure 4.

  1. Give absorbance on Y-axis in Fig. 5B. If absorbance is from 2 to 4 as shown in this Figure this is not correct since at those values there is no linear relationship! The experiments have to be re-done with lover absorbance values!

Response: Thank you for your comments and suggestions. We have carefully examined the raw data and confirmed that the absorbance values are indeed less than 2. The y-axis in the figure should represent 'KDM4 activity,' not the absorbance values. The y-axis has been revised to represent KDM4 activity (OD/min/mg).

We have corrected the unit in Figure 5B from the erroneous one to OD/min/mg to ensure the accuracy of the data. The KDM4 activity was calculated using the following formula:

min: Incubation time (minutes) was 120 min.

 Additionally, we have included a detailed calculation formula in the Materials and Methods 4.13 section to help readers better understand the process of data calculation and unit conversion. 

  1. The obtained values in Figure 6 are sometimes % of input and sometimes Relative mRNA expression and Relative protein expression. Very complicated for a reviewer and reader to grasp what the authors aim to show. The Figure legends and M&M do not help much to interpret these results. This reviewer though wonders whether the authors consider that these minor changes do have an affect on these animals and if so what?

Response: Thanks for the reviewer’s comments and suggestion.

Following the RNA Interference of KDM4, we conducted assessments on the levels of H3K9me3, the H3K9me3 enrichment at PFK and G-6-PD promoters, as well as the mRNA expression of PFK and G-6-PD to investigate the effect of KDM4 expression to the H3K9me3 level and glycolysis level. These data suggest that KDM4 may exert its role in the process of immune priming by modulating the enrichment of H3K9me3 at the promoters of PFK and G-6-PD, thereby potentially influencing glycolytic activity. In addition, we have made the corresponding revisions to the figure legend.

  1. 7C what is mmol/gport on the Y-axis? 

Response: Thanks for the reviewer’s comments. The gprot refer to grams of protein. mmol/gport describing the concentration of lactate in terms of millimoles per gram of protein. The concentration of lactate in the sample was measured to be X millimoles per gram of protein, indicating the amount of lactate produced per unit of protein present. Considering that the unit for lactate concentration might be confusing, it is changed to the international unit mmol/L. In the present study, lactate concentration was determined using a colorimetric assay, and the concentration was calculated using the following formula:

Cpr: protein concentration. Cstandard: standard concentration was 3 mmol/L.

The corresponding modifications have been added to Section 4.12 of the Materials and Methods.

  1. How many animals were used in this experiment in Fig.7? Why are they different measures of concentrations of glucose and lactate in these determination? This makes interpretation of data very difficult. 

Response: Thanks for the reviewer’s comments and suggestion. Twenty-seven crabs were used in this experiment in Fig. 7. The method we chose to measure concentrations of glucose and lactate are applicable for crabs [3]. For glucose, we employed the glucose oxidase method. Regarding lactate, we utilized a lactate dehydrogenase assay. The primary objective of our detection was to illustrate the changes in lactate and glucose content following KDM4 interference. To facilitate the comparison of glucose and lactate concentrations, we have standardized the units to mmol/L.

  1. The ratio of NAD+/NADH is high in EGFP silenced animals and at the same time lactate is high. Is this a plausible correlation?

Response: Thanks for the reviewer’s comments. In the training immunity of vertebrates, the immune cell may exhibit a simultaneous rise in ratio of NAD+/NADH and lactate content [4]. Under steady-state conditions, immune cells have low biosynthetic activity and their energy requirements are predominantly met through oxidative phosphorylation (OXPHOS) and fatty acid oxidation (FAO). Upon activation, the energy demand of innate immune cells increases and aerobic glycolysis, glutaminolysis, cholesterol metabolism and fatty acid synthesis can be used to meet those additional needs [5]. Metabolic intermediates, such as acetyl-CoA, fumarate, succinate, nicotinamide adenine dinucleotide (NAD+) and mevalonate, which are produced as a result of this activation-induced metabolic rewiring, regulate the epigenetic landscape [6]. We have added this discussion in line 260.

We sincerely appreciate the time and effort that the reviewers have invested in evaluating our manuscript. We are eager to receive any additional feedback or suggestions that may further enhance our work.

Sincerely yours,

Lingling Wang

References

  1. Lian, X.; Li, Y.; Wang, W.; Zuo, J.; Yu, T.; Wang, L.; Song, L., The Modification of H3K4me3 Enhanced the Expression of CgTLR3 in Hemocytes to Increase CgIL17-1 Production in the Immune Priming of Crassostrea gigas. Int J Mol Sci 2024, 25, (2).
  2. Yuan, G.; Hu, B.; Ma, J.; Zhang, C.; Xie, H.; Wei, T.; Yang, Y.; Ni, B., Histone lysine methyltransferase SETDB2 suppresses NRF2 to restrict tumor progression and modulates chemotherapy sensitivity in lung adenocarcinoma. Cancer Med 2023, 12, (6), 7258-7272.
  3. Zhu, L.; Qi, S.; Shi, C.; Chen, S.; Ye, Y.; Wang, C.; Mu, C.; Li, R.; Wu, Q.; Wang, X.; Zhou, Y., Optimizing Anesthetic Practices for Mud Crab: A Comparative Study of Clove Oil, MS-222, Ethanol, and Magnesium Chloride. Antioxidants (Basel) 2023, 12, (12).
  4. Cheng, S. C.; Quintin, J.; Cramer, R. A.; Shepardson, K. M.; Saeed, S.; Kumar, V.; Giamarellos-Bourboulis, E. J.; Martens, J. H.; Rao, N. A.; Aghajanirefah, A.; Manjeri, G. R.; Li, Y.; Ifrim, D. C.; Arts, R. J.; van der Veer, B. M.; Deen, P. M.; Logie, C.; O'Neill, L. A.; Willems, P.; van de Veerdonk, F. L.; van der Meer, J. W.; Ng, A.; Joosten, L. A.; Wijmenga, C.; Stunnenberg, H. G.; Xavier, R. J.; Netea, M. G., mTOR- and HIF-1alpha-mediated aerobic glycolysis as metabolic basis for trained immunity. Science 2014, 345, (6204), 1250684.
  5. Acevedo, O. A.; Berrios, R. V.; Rodriguez-Guilarte, L.; Lillo-Dapremont, B.; Kalergis, A. M., Molecular and Cellular Mechanisms Modulating Trained Immunity by Various Cell Types in Response to Pathogen Encounter. Front Immunol 2021, 12, 745332.
  6. Ochando, J.; Mulder, W. J. M.; Madsen, J. C.; Netea, M. G.; Duivenvoorden, R., Trained immunity - basic concepts and contributions to immunopathology. Nat Rev Nephrol 2023, 19, (1), 23-37.

Reviewer 2 Report

Comments and Suggestions for Authors

1)The font is not consistent(i.e. , Figure 2 legend) 2) The context needs further clarification (i.e., no explanation of treatment group in either introduction/results, only hidden in method, very confusing; also never explain the full name of their abbreviation like "NS+NS" , although it is possible to guess that the authors mean negative control) 3) I am not sure why the supposed ladder area of Figure 2A has a black line behind; at least poor quality of experiments 4) no loading control in western blotting; 5) Not  a proper way of testing glycolysis in results 2.1; better use sth like extracellular acidification rate (ECAR) measurements; 6) Never properly explain why test EsKDM4 in the first place; 7) all quantitative figures are not properly labeled, missing individual data(often presented as points) and lower SD(only containing upper SD) in bar charts

Author Response

Dear Reviewer,

Thanks for the thorough review and insightful comments provided on our manuscript. Those comments are all valuable and very helpful for revising and improving our paper, as well as the important guiding significance to our researches. The revised parts are marked with red color in the revised manuscript. Here is a summary of the revisions and responses.

Response to reviewer

  1. The font is not consistent (i.e., Figure 2 legend)

Response: Thanks very much for the reviewer’s suggestion. We have carefully revised the font throughout the manuscript.

  1. The context needs further clarification (i.e., no explanation of treatment group in either introduction/results, only hidden in method, very confusing; also never explain the full name of their abbreviation like "NS+NS", although it is possible to guess that the authors mean negative control)

Response: Thanks for the reviewer’s comments and suggestion. We have revised the figure legends to include detailed explanations of the abbreviations and to provide a more comprehensive understanding of the figures.

  1. I am not sure why the supposed ladder area of Figure 2A has a black line behind; at least poor quality of experiments.

Response: Thanks for the reviewer’s comments. We have repeated the relevant experiments and updated the image in Figure 2A.

  1. no loading control in western blotting;

Response: We appreciate the reviewer’s comments and suggestions. Histone H3 is commonly used as an internal control to correct for sample loaded and potential errors in the experimental process [1]. Since H3K9me3 is directly modified on H3, using H3 as an internal control can more accurately reflect the relative abundance changes of H3K9me3. And we have has been added ‘histone H3 as loading control’ to the figure legends.

  1. Not a proper way of testing glycolysis in results 2.1; better use sth like extracellular acidification rate (ECAR) measurements;

Response: We appreciate the reviewer’s comments and suggestions. Due to there is no report about ECAR detection in crabs yet, we employed the universal approach measurement of lactate and glucose concentrations to evaluate the level of glycolysis using the common method. This method has been widely accepted in the study of invertebrates [2].

  1. Never properly explain why test EsKDM4 in the first place;

Response: We appreciate the reviewer’s comments and suggestions. The KDM4 family of histone demethylases plays a crucial role in regulating chromatin structure and gene expression. KDM4 primarily catalyze the removal of methyl groups from H3K9 and H3K36, thereby affecting the epigenetic state of immune cells. It has been reported that KDM4 mediates the induction of trained immunity by altering H3K9me3 level in vertebrate. In this study, we observed changes in H3K9me3 level at 7 days after first stimulation with A. hydrophila, and speculate that KDM4 may function in A. hydrophila-induced immune priming in crabs. We have added the related information in line 106 and line 183.

  1. All quantitative figures are not properly labeled, missing individual data(often presented as points) and lower SD(only containing upper SD) in bar charts.

Response: We appreciate the reviewer’s comments and suggestions. We have added individual data points (presented as points) to the bar charts. In addition, we have included the lower SD in all bar charts, ensuring that the error bars accurately reflect the range of the data.

We sincerely appreciate the time and effort that the reviewers have invested in evaluating our manuscript. We are eager to receive any additional feedback or suggestions that may further enhance our work.

Sincerely yours,

Lingling Wang

Professor

References

  1. Zhao, S.; Lu, J.; Pan, B.; Fan, H.; Byrum, S. D.; Xu, C.; Kim, A.; Guo, Y.; Kanchi, K. L.; Gong, W.; Sun, T.; Storey, A. J.; Burkholder, N. T.; Mackintosh, S. G.; Kuhlers, P. C.; Edmondson, R. D.; Strahl, B. D.; Diao, Y.; Tackett, A. J.; Raab, J. R.; Cai, L.; Song, J.; Wang, G. G., TNRC18 engages H3K9me3 to mediate silencing of endogenous retrotransposons. Nature 2023, 623, (7987), 633-642.
  2. Zhang, Y.; Zhang, X., Virus-Induced Histone Lactylation Promotes Virus Infection in Crustacean. Adv Sci (Weinh) 2024, 11, (30), e2401017.

Reviewer 3 Report

Comments and Suggestions for Authors

The manuscript by Xinyu Zhao et al describes the regulation of glycolysis in the immune priming of Chinese mitten crab Eriocheir sinensis, focusing on the role of histone demethylase KDM4 in metabolic reprogramming. The manuscript could proceed further after a modification following the comments below.  

Results: Fig 3A is necessary to describe more details both in the main text and legends. Fig 3A, lane 2 shows many protein bands and Fig 3A, lane 3 shows a purified band. The authors need to explain what the bands in Fig 3 are, lane 2. The purified band looks different MW than the bands identified in lane 2. Please explain this issue.

Fig 4: A scale bar for each image is required to be included.  

Methods: The SDS-PAGE experimental procedure is necessary to include in the methodology section.

Discussion: The discussion needs to be modified with relevant updated references. 

Comments on the Quality of English Language

The English could be improved which can help readers to understand and more clearly express the research.

Author Response

Dear Reviewer,

Thanks for the thorough review and insightful comments provided on our manuscript. Those comments are all valuable and very helpful for revising and improving our paper, as well as the important guiding significance to our researches. The revised parts are marked with red color in the revised manuscript. Here is a summary of the revisions and responses.

Response to reviewer

The manuscript by Xinyu Zhao et al describes the regulation of glycolysis in the immune priming of Chinese mitten crab Eriocheir sinensis, focusing on the role of histone demethylase KDM4 in metabolic reprogramming. The manuscript could proceed further after a modification following the comments below.  

  1. Results: Fig 3A is necessary to describe more details both in the main text and legends. Fig 3A, lane 2 shows many protein bands and Fig 3A, lane 3 shows a purified band. The authors need to explain what the bands in Fig 3 are, lane 2. The purified band looks different MW than the bands identified in lane 2. Please explain this issue.

Response: Thanks for the reviewer’s comments and suggestions. We have revised the manuscript to include a more comprehensive description of Figure 3 (including lane 2) in line 159, and updated the figure legend for Figure 3.

It is plausible that the protein loading in lane 2 was excessive, which may lead to band diffusion, thereby affecting the position of the band. This phenomenon also appeared in previous report of our lab [1]. In Fig.3C, western blotting assay the specificity of the polyclonal anti-rEsKDM4 based on the sample of lane 2. The results showed a distinct band, which was in consistence with the predicted molecular mass, suggesting that lanes 2 and 3 in Fig.3A indeed represent rEsKDM4.

  1. Fig 4: A scale bar for each image is required to be included.  

Response: Thanks for the reviewer’s suggestion. We have added a scale bar in each image of Figure 4.

  1. Methods: The SDS-PAGE experimental procedure is necessary to include in the methodology section.

Response: Thanks for the reviewer’s suggestion. The SDS-PAGE experimental procedure has been added to 4.6 in line 432 as follow:

Protein induced and purification were monitored using SDS-PAGE. Bacterial suspension and purified rEsKDM4 in β-mercaptoethanol-containing loading buffer were heated at 100°C for 10 min before loading on SDS-PAGE gels. Protein bands were visualised by staining using Coomassie Protein Stain and were subsequently imaged by Amersham Imager 600 system (GE Healthcare, USA) in Colorimetric mode.

  1. Discussion: The discussion needs to be modified with relevant updated references.

Response: Thanks for the reviewer’s comments and suggestion. We have revised the discussion with recently study, and marked with red color.

We sincerely appreciate the time and effort that the reviewers have invested in evaluating our manuscript. We are eager to receive any additional feedback or suggestions that may further enhance our work.

Sincerely yours,

Lingling Wang

Professor

References

  1. Jiang, D.; Yang, C.; Wang, X.; Ma, X.; He, Z.; Wang, L.; Song, L., The involvement of AMP-activated protein kinase alpha in regulating glycolysis in Yesso scallop Patinopecten yessoensis under high temperature stress. Fish Shellfish Immunol 2023, 140, 108998.

Round 2

Reviewer 1 Report

Comments and Suggestions for Authors

The manuscript is revised according to most of my comments and can be included for publication.

Author Response

Dear  Reviewer,

I would like to extend my sincere gratitude for your positive feedback and for considering our manuscript suitable for publication after revision. Your comments and suggestions have been invaluable in enhancing the quality and clarity of our work.

Thank you once again for your time and for the opportunity to contribute to our work.

Sincerely yours,

Lingling Wang

Professor

Dalian Ocean University, 52 Heishijiao Street, Dalian 116023, China.

E-mail: wanglingling@dlou.edu.cn

Reviewer 2 Report

Comments and Suggestions for Authors

Recommend to accept after minor revisions. I am glad to see the authors have addressed all the problems I mentioned previously. However, there are still some minor problems I detected in your manuscripts. There are possible missing explanations  of Figure 2D and 3A, which could be easily fixed. Authors need to address these problems before acceptance.

Author Response

Dear Editor and Reviewers,

Thanks for the thorough review and insightful comments provided on our manuscript. Those comments are all valuable and very helpful for revising and improving our paper, as well as the important guiding significance to our researches. The revised parts are marked with red color in the revised manuscript. Here is a summary of the revisions and responses.

Response to reviewer

Recommend to accept after minor revisions.  I am glad to see the authors have addressed all the problems I mentioned previously.  However, there are still some minor problems I detected in your manuscripts.  There are possible missing explanations of Figure 2D and 3A, which could be easily fixed.  Authors need to address these problems before acceptance.

Response: We appreciate the reviewer’s comments and suggestions.  We have expanded the results description and figure legend of Figure 2D and Figure 3A in the manuscript to provide a more detailed explanation of the results presented (line 137, line166, line 775-788).

We sincerely appreciate the time and effort that the reviewers have invested in evaluating our manuscript. We hope that with these amendments, the manuscript is now meets the high standards of your expectations.

Sincerely yours,

Lingling Wang

Professor

Dalian Ocean University, 52 Heishijiao Street, Dalian 116023, China.

E-mail: wanglingling@dlou.edu.cn

Reviewer 3 Report

Comments and Suggestions for Authors

The authors addressed all of my comments correctly. I am happy to accept this manuscript in its present form. 

Author Response

(The authors gave the same response as above.)
